# Metal Oxide Compact Electron Transport Layer Modification for Efficient and Stable Perovskite Solar Cells

**DOI:** 10.3390/ma13092207

**Published:** 2020-05-11

**Authors:** Md. Shahiduzzaman, Shoko Fukaya, Ersan Y. Muslih, Liangle Wang, Masahiro Nakano, Md. Akhtaruzzaman, Makoto Karakawa, Kohshin Takahashi, Jean-Michel Nunzi, Tetsuya Taima

**Affiliations:** 1Nanomaterials Research Institute, Kanazawa University, Kakuma, Kanazawa 920-1192, Japan; karakawa@staff.kanazawa-u.ac.jp (M.K.); nunzijm@queensu.ca (J.-M.N.); 2Graduate School of Frontier Science Initiative, Kanazawa University, Kakuma, Kanazawa 920-1192, Japan; sheeepwing-tjkst@stu.kanazawa-u.ac.jp (S.F.); Bro.Wang@outlook.com (L.W.); 3Graduate School of Natural Science and Technology, Kanazawa University, Kakuma, Kanazawa 920-1192, Japan; ersanmuslih@gmail.com (E.Y.M.); masahiro-nakano@se.kanazawa-u.ac.jp (M.N.); ktakaha@se.kanazawa-u.ac.jp (K.T.); 4Solar Energy Research Institute, The National University of Malaysia, Bangi 43600, Malaysia; akhtar@ukm.edu.my; 5Department of Physics, Engineering Physics and Astronomy, Queens University, Kingston, ON K7L-3N6, Canada

**Keywords:** metal oxide, compact layer, electron transport layer, modification layer, perovskite solar cells

## Abstract

Perovskite solar cells (PSCs) have appeared as a promising design for next-generation thin-film photovoltaics because of their cost-efficient fabrication processes and excellent optoelectronic properties. However, PSCs containing a metal oxide compact layer (CL) suffer from poor long-term stability and performance. The quality of the underlying substrate strongly influences the growth of the perovskite layer. In turn, the perovskite film quality directly affects the efficiency and stability of the resultant PSCs. Thus, substrate modification with metal oxide CLs to produce highly efficient and stable PSCs has drawn attention. In this review, metal oxide-based electron transport layers (ETLs) used in PSCs and their systemic modification are reviewed. The roles of ETLs in the design and fabrication of efficient and stable PSCs are also discussed. This review will guide the further development of perovskite films with larger grains, higher crystallinity, and more homogeneous morphology, which correlate to higher stable PSC performance. The challenges and future research directions for PSCs containing compact ETLs are also described with the goal of improving their sustainability to reach new heights of clean energy production.

## 1. Introduction

Energy is one of the fundamental necessities of our current society. At present, the ever-increasing world energy consumption relies on fossil energy resources. Burning fossil fuels releases greenhouse gases, causing global warming, which threatens the Earth’s ecosystems. Therefore, it is strongly desired to explore alternative, carbon-free, renewable energy sources to replace fossil fuels. Solar energy is an attractive alternative energy source because it is the largest usable source that could potentially meet global future energy demands. A solar cell is a device that converts solar energy directly into electrical energy. Solar cells make no noise or pollution and have no moving parts, making them robust, reliable, and long-lasting [1,2,3]. The development of alternative power systems that harvest ambient energy sources (radiant, thermal, and mechanical energy) is desired in order to perpetually power or recharge Internet of Things devices. Solar cells possess several advantageous features, including high stability and a cost-effective fabrication process. Because of these features, solar cells are expected to act as long-term power supplies for satellites and space vehicles [4].

The majority of research on solar cells has focused on silicon-based assemblies, in which silicon is used as both the light absorber and charge transporter [5]. However, the high processing cost, toxicity of silicon-based chemicals, and high energy demands for cell fabrication are disadvantages that currently limit the wide application of silicon-based solar cells. The buildings where such solar cells can be installed are limited. Researchers have been exploring potential next-generation solar cell technologies that are cheap and can be processed using simple solution-based techniques. Solution-based processing has been used to fabricate various types of photovoltaic cells, such as organic photovoltaics, dye-sensitized solar cells, quantum dot solar cells, and perovskite solar cells (PSCs), making these cells attractive lost-cost alternatives to silicon solar cells.

Perovskites are minerals with the same structure as calcium titanium oxide (CaTiO_3_) and are named after the Russian Geographical Society founder Lev Perovski. The light harvester in PSCs is a perovskite such as methylammonium lead iodide (CH_3_NH_3_PbI_3_), with a general formula ABX_3_, where “A” is a univalent organic cation (CH_3_NH_3_^+^), “B” is a bivalent metal ion (Pb^2+^) and “X” is a halide (I^−^) anion (Figure 1a). Here, A coordinates with 12 X anions and B coordinates with six X anions, forming a geometry of cubes and octahedrons, respectively. The crystal structure and perovskite arrangement can be estimated using the Goldschmidt tolerance factor. Hybrid organic–inorganic metal halide PSCs are currently the fastest growing technology in the history of photovoltaics. Such PSCs have attracted broad interest from academic and industrial communities because of their favorable optoelectronic properties, such as tunable bandgap, high absorption coefficient, high electron and hole mobilities, long diffusion lengths (>1 µm), and easy and inexpensive fabrication by solution processing [6,7,8,9]. Intensive research efforts around the world have led to great improvements in the power conversion efficiency (PCE) of PSCs since their original PCE of 3.8% was reported by Miyasaka and co-workers in 2009 [10]. In just over a decade, the PCE of PSCs has rapidly increased to reach the latest record of 25.2%, approaching the top values achieved for single-crystalline silicon solar cells [11]. There are two leading types of device architecture that yield efficient PSCs: planar heterojunctions (PHJs) and mesoporous (mp) structures. Because of their facile device processing and efficient performance, PHJ PSCs have attracted more research interest than mp-PSCs. Typical PSCs have a sandwich structure consisting of a perovskite photoactive layer with a thickness of several hundred nanometers between metal oxide (MO_x_) layers; namely, an electron transport layer (ETL) and hole-transport layer (HTL) (Figure 1b). When a perovskite device is illuminated by a standard simulated solar spectrum, charge carriers are generated within the perovskite photosensitive layer, separated by the ETL and HTL, and collected at their respective electrodes, thereby generating an electric current. A schematic illustration of the energy-level alignment between device components is shown in Figure 1c. The ETL needs to transport photogenerated electrons and block holes to eradicate the electrical shunt and afford stable PSCs with high performance. Figure 1d shows the progress in PSC efficiency by year.

With the rapid improvement of PSC performance, device lifetime (operational stability) has become the main concern for their future commercialization. PSC stability can be affected by both extrinsic (environmental) and intrinsic factors. Environmental factors include moisture, oxygen, light, and heat, which can degrade the photoactivity of the perovskite components [12]. The intrinsic stability of perovskites is strongly influenced by the presence of defects in the perovskite layer and at the interface between the perovskite and charge transport layers [13]. Perovskite instability is dominated by the hygroscopic nature of the organic cations, under-coordinated lead atoms, thermal instability, and ion migration. The PSC degradation driven by the hygroscopic character of the organic cations is correlated with environmental factors, which can be resolved by device encapsulation [14]. Thermal instability can be addressed by compositional engineering of the perovskite and inorganic materials used in the HTL [15,16,17]. Recently, Miyasaka et al. reported the design and fabrication of high-quality perovskite films and dopant-free HTLs for efficient and stable PSCs [18]. Ion migration in perovskite films is another origin of PSC degradation. Ion migration initiated by the illumination-induced electric field can be hindered or even stopped by passivating the grain boundaries in the device; that is, through interfacial modification [19]. Furthermore, ion substitution can decrease the packing density of the crystal lattice [20]. Interfacial modification can also lower the density of recombination hubs correlated with under-coordinated lead atoms or organohalide defect sites, consequently improving the overall PSC performance [21]. Modification of the ETL/perovskite interface plays roles in facilitating charge transport and suppressing hysteresis and interfacial recombination to realize highly efficient and stable PSCs [22]. Therefore, it is immensely desirable to develop facile techniques for the modification of ETL/perovskite interfaces to enable efficient fabrication of low-cost and highly stable PSCs. The present mini-review summarizes recent progress on MO_x_ compact ETLs and their systemic modification to realize high-efficiency and stable PSCs.

## 2. Metal Oxide Compact Electron Transport Layers

MO_x_ is the most uplifting design with respect to the thin-film processing, electronic structure, charge transport mechanisms, defect states, and optoelectronic properties [28]. However, MO_x_ are promising materials in PSCs because they suppress the electrical shunt between the transparent electrode/perovskite and transparent electrode/HTL interfaces as well as permitting electron transport and blocking hole transport to the respective electrode [29,30]. MO_x_ compact electron transport materials such as titanium dioxide (TiO_2_) [31], zinc oxide (ZnO) [32], tin oxide (SnO_2_) [33], zinc stannate (Zn_2_SnO_4_) [34], zirconium dioxide (ZrO_2_) [35], tungsten oxide (WO_3_) [36], and niobium pentoxide (Nb_2_O_5_) [37], can be used as compact ETLs. In particular, efficient PSCs have been fabricated using TiO_2_, SnO_2_ and ZnO films as ETLs.

## 3. Titanium Dioxide (TiO_2_)

The anatase (tetragonal), brookite (orthorhombic), and rutile (tetragonal) polymorphs of TiO_2_ possess distinct crystalline phases and unique properties and have been widely used as photocatalysts [38] and in solar cells [39]. TiO_2_ is the most promising material used in n-type ETLs for efficient PSCs because of its environmentally friendly nature, tunable electronic properties, low cost, and conduction band that is well matched with that of perovskites, thus facilitating electron injection and collection. However, the use of TiO_2_ film in PHJ PSCs has the following drawbacks. (i) The low conductivity and electron mobility of TiO_2_ are unfavorable for electron collection and transport [40,41]. (ii) Exposure of TiO_2_ to ultraviolet light induces the formation of oxygen vacancies at the surface and grain boundaries of TiO_2_ that act as charge traps and result in severe loss of photogenerated carriers through recombination [42,43]. Thus, the interface between TiO_2_ and perovskite retards the photoresponse of the resultant devices and leads to strong hysteresis [41]. Considerable effort has been devoted to improving the performance of PSCs by resolving these drawbacks by modification of TiO_2_ compact layers (CLs) via interface engineering and elemental doping. The surface morphology and properties of the TiO_2_ CL of PSCs strongly affect the quality of the perovskite photosensitive layer in terms of the crystal size, homogeneity, and surface coverage, which in turn influence the photovoltaic performance [44]. The systemic modification of TiO_2_ ETLs is important to optimize the efficiency and stability of PSCs.

### 3.1. TiO_2_ Compact Layers (CLs)

PHJ PSCs with TiO_2_ ETLs have received more research interest than mp-architectures because of their potential to realize low-temperature processing, simple device framework, and high-throughput roll-to-roll manufacturing. Optimization of deposition methods of TiO_2_ CLs is important to achieve uniform compact thickness to maximize the charge transport capacity between the TiO_2_ CL and perovskite. In addition, the thickness of the TiO_2_ CL needs to be optimized to maximize electron transport from the perovskite to the fluoride-doped tin oxide (FTO) electrode. When the TiO_2_ CL is too thick, the resultant increase of the distance of electron transport from the perovskite to FTO lowers the efficiency of charge transport [45]. Conversely, a TiO_2_ CL that is too thin cannot efficiently cover the FTO substrate. Numerous deposition techniques have been used to form TiO_2_ CLs in PSCs, such as the sol–gel method [46], oblique electrostatic inkjet (OEI) deposition, chemical bath deposition (CBD) [47,48], spray pyrolysis [49], atomic layer deposition (ALD) [50], thermal oxidation [31], sputtering [51], chemical vapor deposition (CVD) [52], and electrodeposition [53]. Spin coating is a commonly used simple and robust technique to form TiO_2_ CLs. However, this approach cannot produce high-quality TiO_2_ CLs and has limited suitability for large-scale production. Spray pyrolysis sprays a titanium precursor onto a heated substrate using an atomizer. The precursor droplets thermally decompose to form a TiO_2_ CL. The TiO_2_ CLs produced through either spin coating or spray deposition are particularly sensitive to control parameters. Consequently, the PCE of PSCs may differ considerably even when using the same technique to form TiO_2_ CLs. ALD is a scalable technique for the preparation of TiO_2_ CLs, but is relatively time consuming and expensive. CVD and sputtering require vacuum conditions and have slow deposition rates, which make them expensive and inconvenient for the production of TiO_2_ CLs. Below, we discuss a few commonly used simple and efficient approaches to pattern TiO_2_ CLs for PSCs.

#### 3.1.1. Sol–Gel Technique

The sol–gel approach is widely considered to be a promising technique for the deposition of TiO_2_ CLs because of its simplicity. Segawa and co-workers demonstrated the surface treatment of TiO_2_ CLs using an aqueous solution of TiCl_4_ and ultraviolet/ozone to enhance the wettability and interface adhesion between the TiO_2_ CL and perovskite [54]. The resultant PHJ PSCs showed PCEs of up to 16.9%. Zhou et al. reported planar PSCs with a PCE of 19.3% based on indium tin oxide (ITO)-coated glass treated with ethoxylated polyethyleneimine and a low-temperature TiO_2_ CL [55]. In addition, to increase the conductivity of the TiO_2_ CL, it was doped with yttrium. The favorable band alignment of yttrium-doped TiO_2_ with the perovskite and ITO facilitated efficient electron injection and collection and suppressed interfacial recombination. Choi et al. reported PHJ PSCs with a PCE of 16.3% containing TiO_x_ CLs formed by a simple sol–gel process at room temperature [56]. The performance of these PSCs was comparable to that of cells containing TiO_2_ CLs fabricated at high temperature. Additionally, the room temperature-processed TiO_x_ CL was used as an ETL in a flexible PSC that achieved a PCE of 14.3%. Liu et al. treated low-temperature-processed TiO_2_ CLs with various concentrations of aqueous TiCl_4_ solutions to form ETLs that were uniform, pinhole-free, and had full surface coverage [57]. PSCs containing these ETLs showed efficient electron transport, charge extraction, and suppressed charge recombination, which led to enhanced photovoltaic performance.

#### 3.1.2. Chemical Bath Deposition (CBD)

CBD is a simple, inexpensive, convenient, and scalable technique to pattern TiO_x_ CLs from aqueous solutions, making it advantageous for large-scale production of thin films for organic photovoltaic cells and PSCs [58]. Several reports have been published about CBD TiO_2_-based PSCs. Recently, Liu and co-workers reported PSCs containing co-doped TiO_2_ ETLs formed by low-temperature CBD with a PCE of 19.10% [59]. In addition, the group led by Takahashi and Kuwabara developed a technique to pattern TiO_x_ films at low temperatures and used them as ETLs in organic photovoltaic cells that exhibited a PCE of 3.8% under AM1.5G simulated sunlight [60]. TiO_2_ CLs fabricated by CBD form good physical and electronic contacts with perovskite photosensitive layers. We reported PHJ PSCs with TiO_x_ CLs formed by CBD and modified with a thin layer of fullerene or a derivative as ETLs [47,48,61,62]. The scalable CBD technique solely enables production of TiO_x_ CLs at low temperature [63]. Photographs of the hydrolysis steps of CBD are shown in Figure 2. A 30 nm TiO_x_ film is formed in a single operation. The thickness of TiO_x_ can be increased by repeating the operation. However, it is quite difficult to control the morphology and thickness of TiO_2_ CLs fabricated by CBD, which leads to poor reproducibility. Therefore, it remains important to develop a technique to fabricate high-quality TiO_2_ CLs for efficient PHJ PSCs that is scalable, controllable, and low cost. This process can be used for roll-to-roll production, which may contribute it a potential candidate for low-cost commercial production of TiO_2_-based ETLs in future solar modules.

#### 3.1.3. Inkjet Printing Approaches

Inkjet printing is a scalable technique that is compatible with roll-to-roll processing. In addition, inkjet printing is cost-effective and has a high material utilization rate, which may lower the production cost of devices. Wei et al. reported efficient PSCs containing TiO_2_ CLs formed by electrodeposition [53]. Electrodeposition is a simple, cost-effective, and scalable technique to pattern uniform TiO_2_ CLs. The surface morphology and thickness of TiO_2_ CLs can be controlled by simply manipulating deposition conditions. In addition, we recently developed an OEI bottom-up strategy to produce TiO_2_ CLs that involves discharging a spray using electrostatic force. In the OEI strategy, an FTO substrate was patterned with a TiO_2_ precursor solution using an ejection angle of 45° with respect to the substrate (Figure 3a). The OEI technique produced high-quality TiO_2_ CLs on FTO substrates, which were used as the ETL in PSCs that showed a PCE of 13.19% [64]. Unlike other bottom-up techniques, OEI deposition offers a simple and cost-effective way to obtain high-quality TiO_2_ films with easy-to-control thickness. Using OEI deposition, large-area stacked thin films can be fabricated with high reproducibility. The surface morphology of TiO_2_ films was tuned by simply changing the coating time. Large-scale printing using OEI deposition may be realized by changing the single nozzle system to one with multiple nozzles. OEI deposition can be used to pattern a variety of substrates on a large scale to achieve high-throughput, large-area perovskite solar modules. The scalability, large-area uniformity, and low-cost fabrication capability with a high material utilization rate of OEI deposition might lower the production cost of resultant devices. This simple technique to fabricate TiO_2_ CLs may help to further improve the photovoltaic performance of planar small- and large-scale PSC modules.

### 3.2. Surface Modification by TiO_2_ Nanoparticles

To improve the performance and stability of PSCs, there has been significant focus upon the surface modification of ETLs. TiO_2_ nanoparticles (NPs) possess a higher specific surface area than that of TiO_2_ CLs, enabling the surface morphology of TiO_2_ films to be tuned. TiO_2_ NPs can promote efficient electron injection and transport to improve the balance of charge carriers. The anatase phase of TiO_2_ is extensively used as the ETL in PSCs because of its easy synthesis. Conversely, the brookite phase of TiO_2_ is the least explored because it is difficult to synthesize the pure brookite phase. The rutile phase of TiO_2_ also shows potential for use as an ETL in PSC applications. TiO_2_ CLs composed of mixtures of anatase and rutile phases have been used in PSCs. To explore the performance of different polymorphs of TiO_2_-based PSCs, it is important to understand the effect of TiO_2_ microstructure on PSC performance and stability. At present, inorganic metal oxides (TiO_2_ NPs, mp-TiO_2_) and organics such as self-assembling monolayers (SAM), fullerene (C_60_), [6,6]–phenyl–C_61_–butyric acid methyl ester (PCBM) based materials are used to combine with or modify TiO_2_ CL, SnO_2_, ZnO in PSCs and their corresponding device architectures and PCEs are included in Table 1.

#### 3.2.1. Anatase TiO_2_ Nanoparticles (NPs)

We found that introduction of single-crystalline anatase TiO_2_ NPs, which were synthesized by a hydrothermal approach [76], between the TiO_2_ CL and perovskite layer altered the interface surface morphology to provide highly efficient and stable PSCs [65]. TiO_2_ NPs are water soluble and have a low content of carbon per titanium atom, which make them attractive reagents to prepare titanium-containing functional materials. Large grains with fewer grain boundaries were observed in the perovskite layer when TiO_2_ NPs were introduced between the TiO_2_ CL and perovskite, which facilitated efficient electron extraction and suppressed charge carrier recombination. PSCs containing the TiO_2_ CL/anatase TiO_2_ NP bilayer showed a PCE of 17.05% and decreased hysteresis (Figure 4) compared with that of the equivalent device without TiO_2_ NPs. We also investigated the moisture stability of these bilayer-based PSCs stored a dry, dark N_2_ environment. The PCE of the bilayer-based device showed remarkable long-term moisture stability, deteriorating to 38% of its initial PCE after 47 days. Gan et al. fabricated PSCs with a PCE of up to 3.7% by using TiO_2_ nanorods with a length of 600 nm as an ETL scaffold [77]. He and his colleagues reported efficient PSCs containing an ETL of single-crystalline TiO_2_ nanorods with an average length of 30 ± 10 nm and diameter of 4 ± 1 nm that exhibited a PCE of over 17% [78]. The one-dimensional nanostructure acted solely as a CL in the high-performance PSCs. In other work, high-performance stable PSCs were produced by adding TiO_2_ NPs to a perovskite precursor solution to tune the perovskite morphology [79]. The tuned perovskite morphology improved electron extraction, thereby enhancing the PCE of the PSCs to 19.2%.

#### 3.2.2. Brookite TiO_2_ NPs

Among the three polymorphs of TiO_2_, the brookite phase has been used the least in PSCs because it is difficult to synthesize. However, using brookite TiO_2_ NPs instead of anatase TiO_2_ in PSCs has numerous advantages. Compared with anatase TiO_2_, brookite TiO_2_ has (i) a higher driving force with more favorable energy-level alignment with the perovskite layer, (ii) higher capacity for charge transfer, (iii) increased crystallinity, (iv) synthesis at relatively low temperature, (v) higher conductivity, (vi) higher electron mobility to facilitate photogenerated electron collection, and (vii) higher thermal stability [80,81,82]. The conduction band edge of brookite is slightly higher than those of anatase and rutile TiO_2_, which is expected to promote efficient electron transfer from the brookite to the anatase phase (Figure 5b). The close match of the conduction band edge of brookite with the band structure of perovskites might increase both the open-circuit voltage (*V*_oc_) and efficiency of PSCs. The higher driving force with a more negative flat-band potential and more efficient charge separation and lower resistance of the brookite phase of TiO_2_ with perovskite can be compared with the case for the anatase phase [80]. Charge transport at the semiconductor/electrolyte interface can be influenced by the semiconducting properties of the TiO_2_ phase because the brookite phase shows higher conductivity and thermal stability than the anatase [81]. Miyasaka and co-workers fabricated compact TiO_x_/mp-brookite TiO_2_-based PSCs with a PCE of 21.6% and high *V*_oc_ because of the relative conduction-band level of the brookite layer to that of the perovskite [83]. PSCs with brookite TiO_2_ exhibited a higher *V*_oc_ than those with other phases of TiO_2_. Thus, it is important to explore the performance of PSCs containing ETLs composed of single-phase anatase or brookite and those with anatase-brookite (AB) and brookite-anatase (BA) heterophase junctions to best manipulate the charge transport characteristics of each phase. Recently, we fabricated TiO_2_ heterophase junctions at low temperature (<180 °C) and tested their performance in PSCs. We used different TiO_2_ phases to manipulate the charge transfer in TiO_2_ semiconductor junctions in PSCs (Figure 5d,f). We also found that PSC efficiency can be improved by changing the crystalline phase of the TiO_2_ CL. We showed that PSCs with FTO-AB as an ETL exhibited high efficiency up to 16.82% [66]. In addition, the results suggested that single-phase brookite TiO_2_ NPs without a TiO_2_ CL can act as an effective ETL in PSCs. These PSCs exhibited a PCE of 14.92%, which was comparable to the first work reported by Miyasaka and co-workers [83]. Because of their suitable internecking particles to particles structure, single-phase brookite TiO_2_ NPs can block holes effectively. The large specific surface area of brookite TiO_2_ NPs allows efficient electron injection and subsequent transport that can balance the electron and hole current densities, strongly influencing the performance of PSCs. The rod-like brookite TiO_2_ particles used in these PSCs ranged from 30–50 nm in diameter (Figure 5a). Such particles form a porous scaffold film rather than a CL. These particles cannot be considered as mp-materials because they do not contain pores. According to International Union of Pure and Applied Chemistry (IUPAC) nomenclature, mp-materials possess pores with diameters of 2–50 nm. The performance and stability of PSCs are strongly influenced by the nature of the underlying substrate. Recently, we reported that the incorporation of brookite TiO_2_ NPs as a bridge between the TiO_2_ CL and perovskite layer resulted in PSCs with highly reproducible PCEs of up to ~18.2% with stable performance of 18% under continuous one-sun illumination during maximum power point tracking, light soaking, and ambient atmosphere (Figure 5g–i) [67]. In contrast, the corresponding planar device with only a TiO_2_ CL exhibited a PCE of ~ 15%. Highly stable and conductive brookite TiO_2_ NPs have been used as a bridge between perovskite and TiO_2_ CLs [81]. The NPs facilitated efficient interfacial charge transfer and promoted large grain growth, consequently enhancing device stability and performance. These results indicate that brookite TiO_2_ NPs can be considered an alternative to anatase TiO_2_-based materials in efficient and stable PSCs.

#### 3.2.3. Rutile TiO_2_ NPs

The rutile TiO_2_ phase also shows potential in PSC applications because it is easy to synthesize. Several studies on the application of highly crystalline, oriented rutile TiO_2_ nanorods and nanowires in PSCs have been reported [84,85,86,87]. Park and co-workers reported efficient PSCs containing the rutile TiO_2_ phase [88]. However, rutile TiO_2_ has a more positive conduction band edge potential than that of anatase TiO_2_, resulting in an inferior *V*_oc_ (Figure 6a). In addition, the inset shows more electrons injected to rutile TiO_2_ which can lower the Fermi energy level at equilibrium between E_F_ (TiO_2_) and E_F_ (MAPbI_3_), resulting in the lower *V*_oc_ (Figure 6d). Subsequently, Wu et al. synthesized rutile TiO_2_ NPs by CBD [89]. The NPs were introduced onto FTO substrates for use as the ETL in PSCs, which exhibited a PCE of 15.4% and high reproducibility. Li et al. reported a facile fabrication of TiO_2_ nanorods by a solvothermal method [68]. The nanorods were incorporated in PSCs that exhibited a record PCE of 18.22%. These results demonstrated that the dense and uniform rutile TiO_2_ NPs strongly suppressed hysteresis, which improved the long-term stability of the resultant PSCs. Shao and co-workers grew rutile TiO_2_ thin films on FTO substrates by CBD [90]. PSCs containing these thin films exhibited PCEs of over 12%. This approach is an effective way to produce TiO_2_-based PSCs at low temperature.

### 3.3. Mesoporous TiO_2_

Generally, fabrication of mp-TiO_2_ films requires a complicated and time-consuming process that involves deposition of a TiO_2_ CL followed by synthesis of mp-TiO_2_. In addition to altering the crystalline phase (anatase) of the amorphous oxide film, mp-TiO_2_ requires a high-temperature sintering process at more than 500 °C to improve electron transport properties and remove polymer template molecules. Such a high-temperature, time-consuming process limits the application of mp-TiO_2_ in flexible PSCs fabricated by roll-to-roll processing. Graetzel et al. studied the influence of lithium-doped mp-TiO_2_ on the performance of PSCs [91]. These PSCs showed superior electronic properties because the lithium-doped mp-TiO_2_ lowered electronic trap states, which enabled faster electron transport. Subsequently, Miyasaka and co-workers reported efficient PSCs with a PCE of 21.1% that showed negligible hysteresis by introducing alkali-metal dopants into mp-TiO_2_ (Figure 7) [44]. The doped TiO_2_ films strongly modulated electronic conductivity, which improved charge extraction and hindered charge recombination. In addition, the doped TiO_2_ thin film remarkably influenced the nucleation of the perovskite layer, which subsequently formed large grains that assembled into dense films with facetted crystallites. Huckaba et al. reported PSCs containing inkjet-printed mp-TiO_2_ films that exhibited with a PCE of 18.29% [92]. Overall, inkjet printing technology provides a scalable and reliable alternative to spin coating for large-scale applications. Sung et al. developed PSCs containing mp-TiO_2_ films composed of NPs with a size of 50 nm, which displayed promising performance, including a PCE of 17.19% [93]. Extensive effort has been expended developing nanostructure-based ETL materials for use in PSC applications. Wang et al. revealed that the introduction of a nanocomposite of graphene/TiO_2_ NPs into mp-structured [94]. PSCs led to increased charge collection, which resulted in improved photovoltaic performance. Huang’s group fabricated mp-TiO_2_ nanopillars by room-temperature reactive magnetron sputtering [95]. The nanopillars were then used as the ETL in PSCs. The efficient TiO_2_ CL/mp-TiO_2_ nanopillar scaffold achieved fast carrier extraction and thereby suppressed recombination loss. Several other efficient mp-TiO_2_-based PSCs have also been reported to date [96,97].

### 3.4. Tin Dioxide

SnO_2_ can be considered other potential ETLs widely used in PSCs owing to their favorable optoelectronics properties, such as wide optical bandgap, high electron mobility, high transparency in the visible and near infrared regions and suitable energy level match with perovskites, and simple fabrication of optically transparent dense films by several methods [98]. Miyasaka and colleagues have reported that PSCs with low-temperature processed SnO_2_ as an ETLs exhibited a PCE of 13% with high stable [99]. Hagfeldt et al. have reported a simple chemical bath post-treatment deposited SnO_2_ and used as ETL in PSCs that showed a PCE close to 21% [33]. In addition, You and co-workers have been achieved an impressive planar PSCs with a record PCE of 20.9% by incorporating of thin layer of SnO_2_ nanoparticle as ETLs [100]. In order to further improve the performance of SnO_2_ ETL based PSCs, the elemental doping and surface modification such as surface passivation and bilayer structure techniques have been employed. More importantly, elemental doping with various metal cations such as Li^+^, and Sb^3+^ in SnO_2_ ETLs exhibited efficient planar PSCs [101,102]. Furthermore, Fang and co-workers reported efficient PSCs with a PCE of 18% by using a 3-aminopropyltriethoxysilane self-assembled monolayer to modify the interface between the SnO_2_ and perovskite [103]. The defect passivation technique has been used in SnO_2_-based PSCs by using binary alkaline halides [104]. Yang et al. reported ethylene diaminetetraacetic acid (EDTA) modified SnO_2_, where EDTA facilitates more smooth interface between SnO_2_/perovskite and better match with the conduction band of perovskite, resulting in efficient planar PSCs with a PCE of 21.52% [72]. Chen et al. reported PSCs with a PCE of 13.52% by inserting a simple spin-coating deposited SnO_2_ onto TiO_2_ CL owing to cover cracks of the TiO_2_ hole-blocking layer [105]. Recently, Miyasaka and colleagues reported a stabilize high performance PSCs with a PCE of 22.1%, where TiO_2_ CL was modified by the SnO_2_ layer [70].

### 3.5. Zinc Oxide

ZnO is an outstanding inorganic semiconductor material because it is easy to synthesize, large surface area, and cost-effective fabrication. In addition, ZnO have been studied the most as CLs in PSCs because of their superior optoelectronic properties [106]. Recently, Kim and colleagues reported high-performance and stable PSCs by introducing SAM between ZnO ETL and perovskite in order to improve electron transfer from the perovskite layer to the ETL [32]. Cao et al. demonstrated an effective suppression of the corrosion of perovskite by incorporating the SnO_2_ layer between ZnO and perovskite layers [75]. The PSCs exhibited PCEs as high as 12.17% with minimal hysteresis.

## 4. Other Metal Oxides

Other MO_x_ thin films such as WO_3_ [36,107,108], Zn_2_SnO_4_ [109,110,111,112], Nb_2_O_5_ [37], In_2_O_3_ [113,114], SrTiO_3_ [115,116], BaTiO_3_ [117,118], BaSnO_3_ [119,120], and ZrO_2_ [35], have also been investigated for use as the potential CL in PSCs. The band gap of WO_3_ is smaller compared to TiO_2_, where it shows higher carrier mobility. Furthermore, Amassian and colleagues have demonstrated PSCs with a PCE of 11.24% by inserting a TiCl_4_ layer at the perovskite/WO_3_ interface, implying significantly reduced charge recombination [121]. Modifying the known ETLs by elemental doping and morphology treatment, along with discovering a new MO_x_ with superior properties, are required to further improve the performance and stability of PSCs.

## 5. Summary and Outlook

PSCs are attracting huge interest as next-generation thin-film photovoltaics with the hope of developing economically feasible power supplies to improve our environmental sustainability in the near future. However, several issues that currently hinder the commercialization of PSCs need to be considered, particularly their efficiency and stability. TiO_2_ thin films have been widely used as ETLs in PSCs because of their excellent electrical properties, high durability, suitable energy-level match with perovskites, and easy fabrication of optically transparent dense films by several techniques. Recently, numerous attempts to improve device performance and stability through interface modification of TiO_2_ CLs with different phases of TiO_2_ NPs, elemental doping, and solvent engineering have been explored. Ideal ETL modification needs to be simple, conducted at low temperature, cost-effective, and provide an ETL with high thermal and chemical stability, suitable band alignment with the perovskite, and excellent wettability with perovskite precursor solution to yield efficient and stable PSCs suitable for future commercialization. From this perspective, we reviewed the progress of bilayer ETLs, which have improved both the performance and stability of PSCs.

To further improve the operational stability of PSCs, simultaneous device engineering such as interface modification of TiO_2_ CL/perovskite, tuning of the crystallinity, morphology, and compositional engineering (both cations/halogens) of the perovskite along with optimization of the organic/inorganic HTL should be considered to realize their full potential. In addition, techniques suitable for large-area fabrication at low cost are highly desired to enable the development of large-area PSC modules. Further optimization of Pb-free perovskites may improve their sustainability to reach new heights in clean energy generation. Experiments should consider both the stability and efficiency of Pb-based and Pb-free PSCs to assist the solar energy revolution.

## Figures and Tables

**Figure 1 materials-13-02207-f001:**
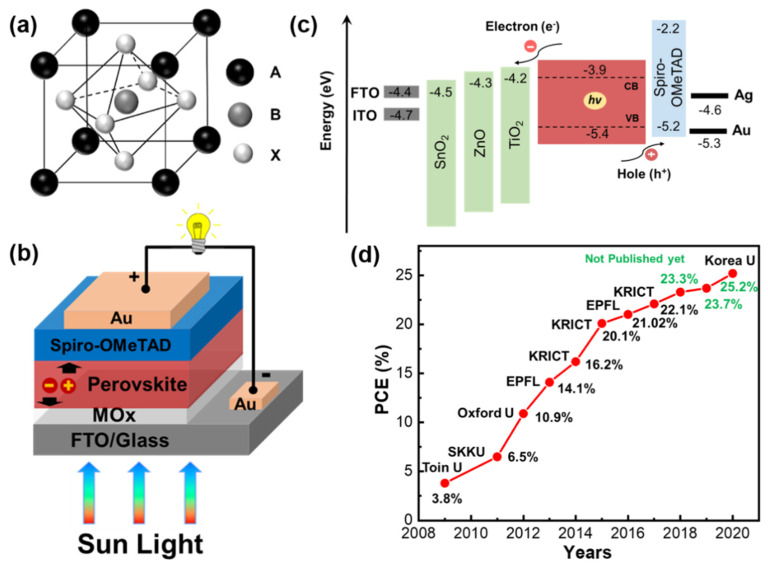
(**a**) General structure of a perovskite, (**b**) schematic of a perovskite solar cell with various layers, (**c**) energy band alignment of MAPbI_3_ perovskite and various metal oxide electron transport materials in the device, and (**d**) performances evolution of perovskite solar cells certified by National Renewable Energy Laboratory (NREL) (14.1% [23], 16.2% [24], 20.1% [25], 21.02% [26] and 22.1% [27]). The green power conversion efficiencies (PCEs, 23.3%, 23.7%, and 25.2%) are not in the public domain yet. SKKU, EPFL, and KRICT denote Sungkyunkwan University, École polytechnique fédérale de Lausanne, and Korea Research Institute of Chemical Technology, respectively.

**Figure 2 materials-13-02207-f002:**
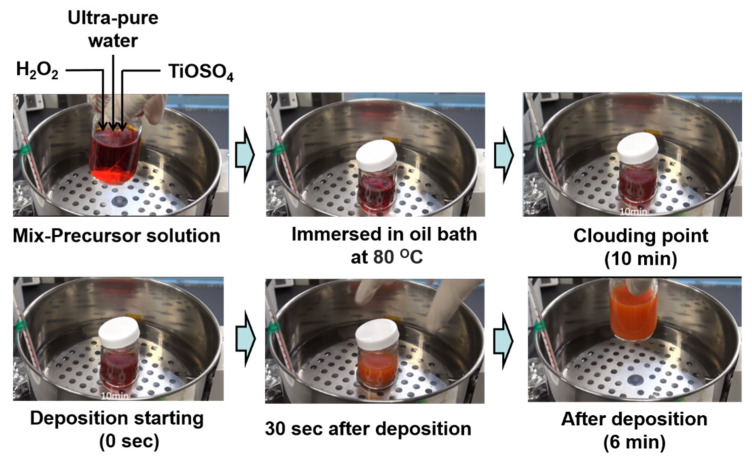
Photographs illustrating the hydrolysis steps in chemical bath deposition of TiO_x_ films.

**Figure 3 materials-13-02207-f003:**
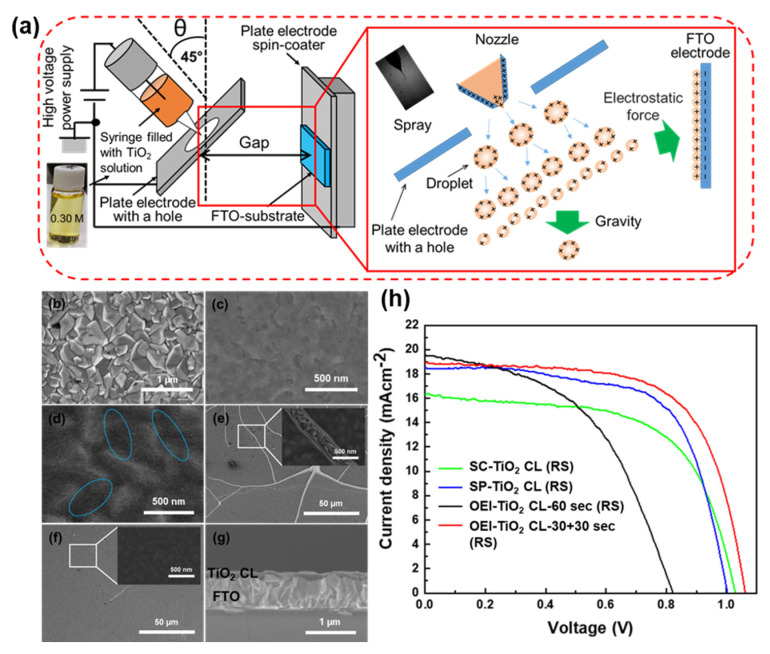
(**a**) Schematic of the oblique electrostatic inkjet (OEI) setup used to pattern TiO_2_ compact layers (CLs) on fluoride-doped tin oxide (FTO)-coated glass substrates. Top-view scanning electron microscope (SEM) images of (**b**) bare FTO, (**c**) spin coating (SC)-TiO_2_ CL, (**d**) spray pyrolysis (SP)-TiO_2_ CL, (**e**) OEI-TiO_2_ CL-60 s, and (**f**) OEI-TiO_2_ CL-30 + 30 s. (**g**) Cross-sectional SEM image of the OEI-TiO_2_ CL-30 + 30 s film. (**h**) Reverse *J-V* characteristics of perovskite solar cells (PSCs) made under different conditions. Reproduced with permission [64]. Copyright 2019, Springer Nature.

**Figure 4 materials-13-02207-f004:**
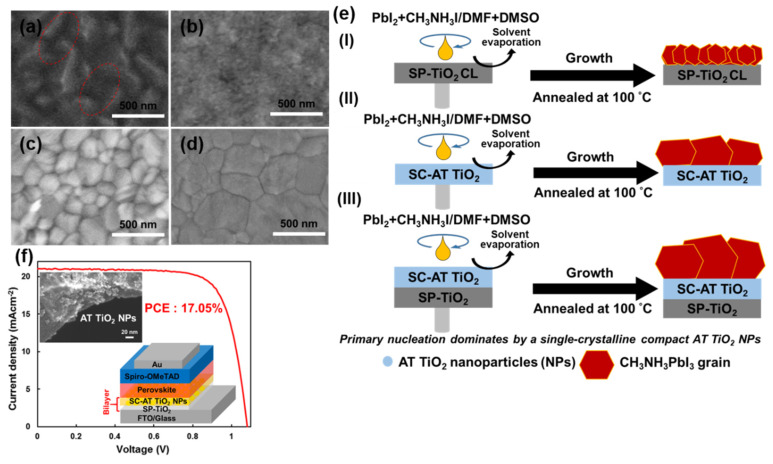
SEM images of the (**a**) SP-TiO_2_ layer and (**b**) SP-TiO_2_/SC-anatase (AT) TiO_2_ nanoparticle (NP) bilayer and perovskite films grown on the (**c**) SP-TiO_2_ layer and (**d**) SP-TiO_2_/SC-AT TiO_2_ NP bilayer. (**e**) Schematic of the nucleation and growth of perovskite grains. (**f**) Reverse *J-V* curve obtained for a solar cell with the NP bilayer. Insets shows the device structure and an SEM image of the NPs. Reproduced with permission [65]. Copyright 2018 American Chemical Society.

**Figure 5 materials-13-02207-f005:**
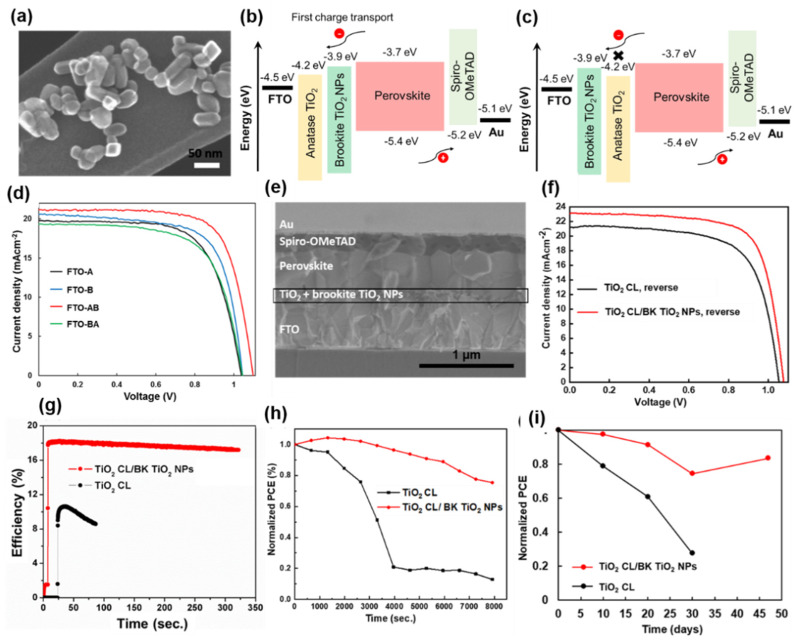
(**a**) Scanning transmission electron microscope (STEM) image of brookite (BK) TiO_2_ NPs. Schematic illustrations of the energy-level alignment in devices with (**b**) FTO-anatase-brookite (AB) and (**c**) FTO-brookite-anatase (BA) as electron transport layers (ETLs). (**d**) *J-V* curves of PSCs with FTO-anatase (A), FTO-brookite (B), FTO-AB, and FTO-BA as ETLs. (**e**) Cross-sectional SEM image of the PSC with an FTO-AB ETL. (**f**) Reverse *J-V* curves obtained for the PSCs on substrates with and without a brookite (BK) TiO_2_ NP layer. (**g**) Efficiency at maximum power point under continuous one-sun illumination (100 mW/cm^2^). Reproduced with permission [66]. Copyright 2019 American Chemical Society. (**h**) Normalized PCE vs. time (in seconds) of TiO_2_ CL/BK TiO_2_ NP-based PSCs (non-encapsulated) after continuous exposure to one-sun illumination for 2.12 h in air with 60–70% relative humidity. (**i**) Normalized PCE versus time (in days) of PSCs (non-encapsulated) stored in the dark under dry N_2_. The devices were measured under ambient conditions at 60–70% relative humidity. Reproduced with permission [67]. Copyright 2020, Royal Society of Chemistry.

**Figure 6 materials-13-02207-f006:**
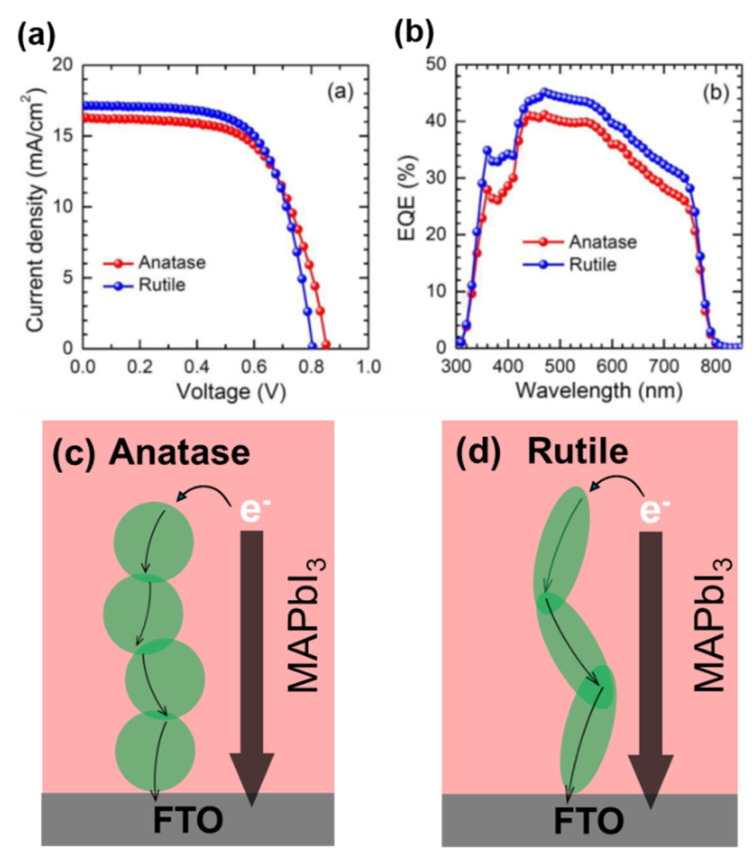
(**a**) *J-V* curves and (**b**) incident photon-to-current efficiency (IPCE) spectra of devices containing anatase and rutile TiO_2_ thin films. Schematic of electronic behavior of devices with (**c**) anatase TiO_2_ and (**d**) rutile TiO_2_ films. Reproduced with permission [88]. Copyright 2014, Royal Society of Chemistry.

**Figure 7 materials-13-02207-f007:**
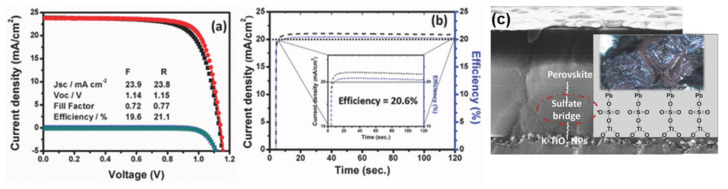
(**a**) *J-V* characteristics of the best-performing PSC fabricated on K2 substrates, which showed a PCE of 21.1% and *V*_OC_ of 1.15 V. (**b**) Steady-state PCE and *J_sc_* at maximum power point of the best-performing device. Inset is a magnified image of the efficiency (blue dotted line) and *J*sc (black dotted line)). (**c**) Combined cross-sectional SEM image and schematic of a doped TiO_2_ mesoporous layer and the surface states bonding with the ETL and perovskite. Reproduced with permission [44]. Copyright 2018, Wiley-VCH Verlag GmbH & Co. KGaA, Weinheim.

**Table 1 materials-13-02207-t001:** Examples of surface modification of TiO_2_, SnO_2_ and ZnO CL in PSCs. The corresponding device architectures and PCEs are derived.

Surface Modification	Devices Structure	PCE (%)	Ref.
TiO_2_/anatase TiO_2_ NPs	FTO/TiO_2_/anatase TiO_2_ NPs/MAPbI_3_/Spiro/Au	17.1	[65]
TiO_2_/brookite TiO_2_ NPs	FTO/TiO_2_/brookite TiO_2_NPs/Cs_0.05_(FA_0.83_MA_0.17_)_0.95_Pb(I_0.83_Br_0.17_)_3_/Spiro/Au	16.8	[66]
TiO_2_/brookite TiO_2_ NPs	FTO/TiO_2_/brookite TiO_2_NPs/MAPbI_3_/Spiro/Au	18.2	[67]
TiO_2_/mp-TiO_2_	FTO/TiO_2_/mp-TiO_2_/Cs_0.05_(FA_0.83_MA_0.17_)_0.95_Pb(I_0.83_Br_0.17_)_3_/Spiro/Au	21.1	[44]
TiO_2_/rutile TiO_2_ NRs	FTO/TiO_2_/rutile TiO_2_ NPs/MAPbI_3_/Spiro/Ag	18.2	[68]
TiO_2_/C60	ITO/TiO_2_/C60/MAPbI_3_/Spiro/Ag	9.5	[47]
TiO_2_/PDI-glass	ITO/TiO_2_/PDI-glass/MAPbI_3_/Spiro/Au	5.7	[61]
TiO_2_/PNP	ITO/TiO_2_/PNP/MAPbI_3_/Spiro/Ag	8.2	[48]
TiO_2_/SnO_2_	FTO/TiO_2_/SnO_2_/MAPbI_3_/PATT/Au	19.8	[69]
TiO_2_/SnO_2_	FTO/TiO_2_/SnO_2_/FA_0.83_MA_0.17_(Ge_0.03_Pb_0.97_(I_0.9_Br_0.1_)_3_/Spiro/Au	22.1	[70]
TiO_2_/SnO_2_	FTO/TiO_2_/SnO_2_/MAPbI_3_/Spiro/Ag	21.1	[71]
TiO_2_/SnO_2_	ITO/TiO_2_/SnO_2_/FA_0.95_Cs_0.05_PbI_3_/PCBM/Ag	21.5	[72]
SnO_2_/SAM	FTO/SnO_2_/SAM/MAPbI_3_/Spiro/Au	18.8	[73]
SnO_2_/PCBM	FTO/SnO_2_/PCBM/MAPbI_3_/Spiro/Au	19.1	[74]
ZnO/SnO_2_	FTO/SnO_2_/PCBM/MAPbI_3_/Spiro/Ag	12.17	[75]
ZnO/SAM	ITO/ZnO/SAM/MAPbI_3_/Spiro	13.7	[32]

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
