# Peer review of "Metal Oxide Compact Electron Transport Layer Modification for Efficient and Stable Perovskite Solar Cells"

_materials, 2020, doi:10.3390/ma13092207_

Round 1

Reviewer 1 Report

The manuscript “Metal Oxide Compact Electron Transport Layer Modification for Efficient and Stable Perovskite Solar Cells” coauthored by Shahiduzzaman et al. had been carefully reviewed. The manuscript is a review paper and the authors claim that “In this review, metal-oxide-based electron transport layers (ETLs) used in Perovskite solar cells (PSCs) and their systemic modification are reviewed”. In recent years, Perovskite Solar Cells related research has received extensive attention. Therefore, I agree that this article has its importance and will attract the interest of readers. However, I have a few suggestions for the content of the article as follows.

1) The title of this article is "Metal Oxide...". However, this article spends a lot of content reviewing TiO2, but only a little for other metal oxides. Therefore, it is suggested that the authors modify the title to focus on TiO2 or supplement the content of the article.

2) In order to allow readers to easily grasp the important information of the reviewed articles. It is recommended that the authors summarize the important information of the reviewed articles into a table, such as publishing year, authors, material, method, conversion efficiency, etc.

3) The Figure 1d has not been explained in the manuscript. Authors are encouraged to compile the relevant information of the PSCs papers in each year in Figure 1d into a table to help illustrate Figure 1d.

4) The manuscript needs to be carefully revised and reorganized.

Author Response

REVISION NOTES

Thank you very much for your letter of April 27, 2020, with regard to our review manuscript (materials-788517; Title: Metal Oxide Compact Electron Transport Layer Modification for
Efficient and Stable Perovskite Solar Cells
) together with the comments. We cordially accept major revision decision for our manuscript (materials-788517). Answers to the specific comments are also provided. Our incorporations of the reviewer suggestions are as follows:

Reply to the Reviewer Comments

Reviewer #1:

The manuscript “Metal Oxide Compact Electron Transport Layer Modification for Efficient and Stable Perovskite Solar Cells” coauthored by Shahiduzzaman et al. had been carefully reviewed. The manuscript is a review paper and the authors claim that “In this review, metal-oxide-based electron transport layers used in Perovskite solar cells and their systemic modification are reviewed”. In recent years, Perovskite Solar Cells related research has received extensive attention. Therefore, I agree that this article has its importance and will attract the interest of readers. However, I have a few suggestions for the content of the article as follows.

  1. The title of this article is "Metal Oxide...". However, this article spends a lot of content reviewing TiO2, but only a little for other metal oxides. Therefore, it is suggested that the authors modify the title to focus on TiO2or supplement the content of the article.

Thank you very much for your opinion. We apologized for the misprint in the information and resulted arises in the confusion. We have expanded the contents in the main body and revised the entire manuscript, implying that the current title shows the consistent in accordance with the content in the main body of the manuscript. Thus, we kept the title of our manuscript as it is. Thank you.

  1. In order to allow readers to easily grasp the important information of the reviewed articles. It is recommended that the authors summarize the important information of the reviewed articles into a table, such as publishing year, authors, material, method, conversion efficiency, etc.

Thank you very much for your precise observation. According to your suggestive notes, we have summarized the important information of the reviewed articles into a Table. Please see the Table No. 1. Hope that the revised version will produce a more balanced and better account of our work, suggesting that the readers will be able to grasp easily the important information of the reviewed articles.  Thank you.

  1. The Figure 1d has not been explained in the manuscript. Authors are encouraged to compile the relevant information of the PSCs papers in each year in Figure 1d into a table to help illustrate Figure 1d.

Thank you very much. We are extremely sorry for this. However, we have added the performances evolution of perovskite solar cells by year. Please see the Figure 1d. Hope that the honorable reviewer will be happy with this. Thank you.

  1. The manuscript needs to be carefully revised and reorganized.

Thank you very much. According to your suggestive notes, we have further checked the claims and revised it in the manuscript. Please see the revision marked version to track all the changes that we have incorporated in the revised manuscript. We hope, the revised manuscript will help the readers to understand our work more precisely and it will be suitable for publication in Materials. Thank you.

We would like to thank the reviewers for his/her helpful comments and hope that the revised version will produce a more balanced and better account of our work. We believe the manuscript has been improved satisfactorily and hope it will be accepted for publication in the Journal of Materials.

Sincerely yours,

Dr. Md. Shahiduzzaman

Nanomaterials Research Institute, Kanazawa University, Japan

Reviewer 2 Report

This manuscript is a review that deals with the preparation of Perovskite Solar Cells (PSCs) as promising materials for photovoltaics and energy production due to their optoelectronic features. In particular, some limitations in the use of a PSCs containing a compact layer of metal oxides need to be taken into account. Hence, the design of metal-oxide based electron transport layers is reported together with some physico-chemical modifications in order to get higher performances. Besides, authors deeply focused on the use of TiO2 in its several crystalline phases, like nanoparticles, rutile, anatase, brookite or mesoporous. The topic of this review is of wide interest and it falls within the scope of the journal. Therefore, I suggest the manuscript to be accepted and published after a few minor revisions:

  • I suggest to modify letters nomenclature in fig 4e without using a-c again, because it could confuse readers.
  • I also suggest to make all the graphs layout uniform throughout the whole MS, with same font text in axes description and same lines thickness, when possible. 

Finally, I suggest also to report the use of other metals in the introduction and to expand the literature (e.g. Nonprecious Copper‐Based Transparent Top Electrode via Seed Layer–Assisted Thermal Evaporation for High‐Performance Semitransparent n‐i‐p Perovskite Solar Cells https://doi.org/10.1002/admt.201800688 )

Author Response

REVISION NOTES

Thank you very much for your letter of April 27, 2020, with regard to our review manuscript (materials-788517; Title: Metal Oxide Compact Electron Transport Layer Modification for
Efficient and Stable Perovskite Solar Cells
) together with the comments. We cordially accept major revision decision for our manuscript (materials-788517). Answers to the specific comments are also provided. Our incorporations of the reviewer suggestions are as follows:

Reply to the Reviewer Comments

Reviewer #2:

This manuscript is a review that deals with the preparation of Perovskite Solar Cells (PSCs) as promising materials for photovoltaics and energy production due to their optoelectronic features. In particular, some limitations in the use of a PSCs containing a compact layer of metal oxides need to be taken into account. Hence, the design of metal-oxide based electron transport layers is reported together with some physico-chemical modifications in order to get higher performances. Besides, authors deeply focused on the use of TiO2 in its several crystalline phases, like nanoparticles, rutile, anatase, brookite or mesoporous. The topic of this review is of wide interest and it falls within the scope of the journal. Therefore, I suggest the manuscript to be accepted and published after a few minor revisions:

  1. I suggest to modify letters nomenclature in fig 4e without using a-c again, because it could confuse readers.

Thank you very much for your precise observation. According to your suggestive notes, we have further modified the claims and revised it in the manuscript. Please see in Figure 4e. Thank you.

  1. I also suggest to make all the graphs layout uniform throughout the whole MS, with same font text in axes description and same lines thickness, when possible.

Thank you very much. We have checked the entire manuscript and revised the whole MS with the same front text in the manuscript. Please see the revision marked version to track all the changes that we have incorporated in the revised manuscript. Thank you.

  1. Finally, I suggest also to report the use of other metals in the introduction and to expand the literature (e.g. Nonprecious Copper‐Based Transparent Top Electrode via Seed Layer–Assisted Thermal Evaporation for High‐Performance Semitransparent n‐i‐p Perovskite Solar Cells https://doi.org/10.1002/admt.201800688).

Thank you very much. We have added a new reference (Adv. Mater. Technol. 2019, 4, 1800688) in the revised manuscript. However, we hope, the revised manuscript will help the readers to understand our work more precisely and it will be suitable for publication in Materials. Thank you.

We would like to thank the reviewers for his/her helpful comments and hope that the revised version will produce a more balanced and better account of our work. We believe the manuscript has been improved satisfactorily and hope it will be accepted for publication in the Journal of Materials.

Sincerely yours,

Dr. Md. Shahiduzzaman

Nanomaterials Research Institute, Kanazawa University, Japan

Reviewer 3 Report

Shahiduzzaman et. al present a review on the recent progress of metal oxide based electron transporting layer for efficient and stable perovskite solar cells. However, the content presented in this manuscript merely review the TiO2 based electron transporting layer, which is much less than the expectation according to the title “Metal Oxide Compact Electron Transport Layer Modification for Efficient and Stable Perovskite Solar Cells”.

  1. Besides TiO2, the authors should review more about other metal oxide based compact layer and their useful modification, such as metal doping, surface modification and etc., rather than presenting a list of metal oxide based electron transporting layer in lines 375-378. The content of that paragraph should be expended to present us more about the recent progress of metal oxide compact layer and related modification for stable and high efficiency perovskite solar cells.
  2. For the content of TiO2 based electron transporting layer, the authors majorly present the recent techniques of the compact layer fabrication and the further modification by various TiO2 Based on our understanding, there are more modification approaches to the TiO2 compact layer for boosting the efficiency and stability of perovskite solar cells, such as surface morphology modification, metal doping, additional organic layer modification, additional inorganic layer modification, and etc. The author should present more about the modification of the compact layer.
  3. The title should be revised to be in accordance with the content in the main body of the manuscript.

Author Response

REVISION NOTES

Thank you very much for your letter of April 27, 2020, with regard to our review manuscript (materials-788517; Title: Metal Oxide Compact Electron Transport Layer Modification for
Efficient and Stable Perovskite Solar Cells
) together with the comments. We cordially accept major revision decision for our manuscript (materials-788517). Answers to the specific comments are also provided. Our incorporations of the reviewer suggestions are as follows:

Reply to the Reviewer Comments

Reviewer #3:

Shahiduzzaman et. al present a review on the recent progress of metal oxide based electron transporting layer for efficient and stable perovskite solar cells. However, the content presented in this manuscript merely review the TiO2 based electron transporting layer, which is much less than the expectation according to the title “Metal Oxide Compact Electron Transport Layer Modification for Efficient and Stable Perovskite Solar Cells”.

  1. Besides TiO2, the authors should review more about other metal oxide based compact layer and their useful modification, such as metal doping, surface modification and etc., rather than presenting a list of metal oxide based electron transporting layer in lines 375-378. The content of that paragraph should be expended to present us more about the recent progress of metal oxide compact layer and related modification for stable and high efficiency perovskite solar cells.

Thank you very much for your opinion. According to your suggestive notes, we have summarized the surface modification of TiO2, SnO2 and ZnO compact layer in PSCs and corresponding reviewed articles into a Table. Please see the Table No. 1. In addition to this, we have expanded the content of metal oxides such as SnO2 and ZnO compact layers and related modification for stable and high efficiency perovskite solar cells. Please see the page number from 377 to 407. Thank you.

  1. For the content of TiO2based electron transporting layer, the authors majorly present the recent techniques of the compact layer fabrication and the further modification by various TiO2Based on our understanding, there are more modification approaches to the TiO2compact layer for boosting the efficiency and stability of perovskite solar cells, such as surface morphology modification, metal doping, additional organic layer modification, additional inorganic layer modification, and etc. The author should present more about the modification of the compact layer.

Thank you. Please see the revision marked version to track all the changes that we have incorporated in the revised manuscript. We hope, the revised manuscript will help the readers to understand our work more precisely and it will be suitable for publication in Materials. Thank you.

  1. The title should be revised to be in accordance with the content in the main body of the manuscript.

Thank you. We have expanded the contents in the main body and revised the entire manuscript, implying that the current title shows consistent in accordance with the content in the main body of the manuscript. Thus, we kept the title of our manuscript as it is. Thank you.

We would like to thank the reviewers for his/her helpful comments and hope that the revised version will produce a more balanced and better account of our work. We believe the manuscript has been improved satisfactorily and hope it will be accepted for publication in the Journal of Materials.

Sincerely yours,

Dr. Md. Shahiduzzaman

Nanomaterials Research Institute, Kanazawa University, Japan

Round 2

Reviewer 1 Report

The revised manuscript had been carefully reviewed. The content shows that the manuscript has been significantly improved and now warrants publication in Materials.

Reviewer 3 Report

The revised version has addressed most of my concerns and can be accepted for the publication.